# Meta-Research: Understudied genes are lost in a leaky pipeline between genome-wide assays and reporting of results

**Reese Richardson[1,2], Heliodoro Tejedor Navarro[2,3], Luis A Nunes Amaral[2,3,4,5]\*, Thomas Stoeger[2,6,7]\***

[1]Interdisciplinary Biological Sciences, Northwestern University, Evanston, United States; [2]Department of Chemical and Biological Engineering, Northwestern University, Evanston, United States; [3]Northwestern Institute on Complex Systems, Northwestern University, Evanston, United States; [4]Department of Molecular Biosciences, Northwestern University, Evanston, United States; [5]Department of Physics and Astronomy, Northwestern University, Evanston, United States; [6]The Potocsnak Longevity Institute, Northwestern University, Chicago, United States; [7]Simpson Querrey Lung Institute for Translational Science, Northwestern University, Chicago, United States

**\*For correspondence:**
amaral@northwestern.edu (LAN);
thomas.stoeger@northwestern.edu (TS)

**Competing interest:** The authors declare that no competing interests exist.

**Abstract** Present-day publications on human genes primarily feature genes that already appeared in many publications prior to completion of the Human Genome Project in 2003. These patterns persist despite the subsequent adoption of high-throughput technologies, which routinely identify novel genes associated with biological processes and disease. Although several hypotheses for bias in the selection of genes as research targets have been proposed, their explanatory powers have not yet been compared. Our analysis suggests that understudied genes are systematically abandoned in favor of better-studied genes between the completion of -omics experiments and the reporting of results. Understudied genes remain abandoned by studies that cite these -omics experiments. Conversely, we find that publications on understudied genes may even accrue a greater number of citations. Among 45 biological and experimental factors previously proposed to affect which genes are being studied, we find that 33 are significantly associated with the choice of hit genes presented in titles and abstracts of -omics studies. To promote the investigation of understudied genes, we condense our insights into a tool, *find my understudied genes* (FMUG), that allows scientists to engage with potential bias during the selection of hits. We demonstrate the utility of FMUG through the identification of genes that remain understudied in vertebrate aging. FMUG is developed in Flutter and is available for download at fmug.amaral.northwestern.edu as a MacOS/Windows app.

## eLife assessment

This study investigated the factors related to understudied genes in biomedical research. It showed that understudied genes are largely abandoned at the writing stage, and it identified a number of biological and experimental factors that influence which genes are selected for investigation. The study is an **important** contribution to this branch of meta-research, and the evidence in support of the findings is **solid**.

## Introduction

Research into human genes concentrates on a subset of genes that were already frequently investigated prior to the completion of the Human Genome Project in 2003 (*Hoffmann and Valencia, 2003*; *Su and Hogenesch, 2007*; *Edwards et al., 2011*; *Gillis and Pavlidis, 2013*; *Stoeger and Nunes*

**eLife digest** Modern techniques for studying human genetics have helped to identify 20,000 protein-encoding genes in the human genome. Yet scientists have not studied most of them, including genes linked to human diseases in genome wide studies. For example, about 44% of the genes associated with Alzheimer's disease have never been mentioned in the title or summary of a scientific article. Why so many health-linked genes have yet to be examined is unclear.

Many genetic studies instead focus on genes already studied before the Human Genome Project mapped the entire genome in 2003. There are many reasons why scientists may ignore potentially disease-causing genes. They may feel that well-studied genes are safer bets or more likely to result in high-profile publications. Or they may lack the tools to study less well-characterized genes.

Richardson et al. analyzed the scientific literature for clues on why so many genes are being ignored by scientists. The analysis included hundreds of articles that used a wide range of genetic techniques, including genome-wide association studies, RNA sequencing, and gene editing tools to scour the genome for disease-linked genes. It revealed that scientists abandon the study of many genes early in the research process and identify 33 reasons why. Contrary to scientists' fears, Richardson et al. show that reports on understudied genes often garner more attention than studies on well-known genes.

Richardson et al. used their results to create a downloadable tool called "Find My Understudied Genes (FMUG)" to help scientists identify understudied genes and counteract bias toward more well-studied genes. The app may help scientists make informed decisions about which understudied genes to research. If the tool helps boost investigation of understudied genes, it may help speed up progress towards understanding human genetics and how various genes may contribute to diseases.

*Amaral, 2022*). This concentration stems from historically acquired research patterns rather than present-day experimental possibilities (*Grueneberg et al., 2008*; *Stoeger et al., 2018*). For most human diseases, these patterns lead to little correlation between the volume of literature published on individual genes and the strength of supporting evidence from genome-wide approaches (*Riba et al., 2016*; *Haynes et al., 2018*; *Border et al., 2019*; *Stoeger and Nunes Amaral, 2020*; *Zhang et al., 2020*; *Byrne et al., 2022*). For instance, we found that 44% of the genes identified as promising Alzheimer's disease targets by the U.S. National Institutes of Health (NIH) Accelerating Medicine Partnership for Alzheimer's Disease (AMP-AD) initiative have never appeared in the title or abstract of any publication on Alzheimer's disease (*Byrne et al., 2022*). Furthermore, when comparing gene-disease pairs, there is no correlation between the ranks of support by transcriptomics and occurrence in annotation databases (*Haynes et al., 2018*).

Although -omics technologies can provide insights on numerous genes across the genome at a time and thus offer the promise to counter historically acquired research patterns (*Collins et al., 2003*; *Shendure et al., 2019*; *Lloyd et al., 2020*; *Kustatscher et al., 2022*), this discrepancy has persisted (*Haynes et al., 2018*; *Rodriguez-Esteban and Jiang, 2017*; *Oprea et al., 2018*; *Sinha et al., 2018*; *Wood et al., 2019*; *Donohue and Love, 2024*) even as the popularity of -omics technologies has risen (*Stoeger and Nunes Amaral, 2022*; *Peña-Castillo and Hughes, 2007*; *Ellens et al., 2017*). We therefore sought to use bibliometric data to delineate where and why understudied human protein-coding genes are abandoned as research targets following -omics experiments. In the absence of any prior quantitative testing of existing hypotheses, it remains unclear whether policies to promote the exploration of a greater set of disease-related genes should focus on how experiments are conducted, how results are reported, or how these results are subsequently received by other scientists.

## Data

We considered 450 genome-wide association studies (GWAS, from studies indexed by the NHGRI-EBI GWAS catalog [*Buniello et al., 2019*]), 296 studies using affinity purification–mass spectrometry (AP-MS, indexed by BioGRID [*Oughtred et al., 2021*]), 148 transcriptomic studies (indexed by the EBI Gene Expression Atlas, EBI-GXA [*Papatheodorou et al., 2018*]), and 15 genome-wide screens using CRISPR (indexed by BioGRID Open Repository of CRISPR Screens, BioGRID ORCS [*Oughtred et al., 2021*]) (see PRISMA diagrams in *Figure 1—figure supplement 1*, *Figure 1—figure supplement 2*, *Figure 1—figure supplement 3*, and *Figure 1—figure supplement 4*). We denote genes that are

found to have statistically significant changes in expression or associations with a phenotype as 'hit' genes.

As a surrogate for a given gene having been investigated closer, we consider whether it was reported in the title or abstract of a research article. We determined which genes were mentioned in the title or abstract of articles using annotations from gene2pubmed (*Maglott et al., 2007*) and PubTator (*Wei et al., 2019*). We used NIH iCite v32 for citations (*Hutchins and Santangelo, 2019*). For determining which gene properties were associated with selection as research targets, we synthesized quantitative measures from a variety of authoritative sources (see Materials and methods).

## Results
### Understudied genes are abandoned at synthesis/writing stage

We sought to identify at which point in the scientific process understudied genes are ignored as research targets in investigations using -omics experiments (*Figure 1A*). To receive scholarly attention, a gene must travel through a pipeline from biological reality to experimental results to write-up of those results. These results must be extended by subsequent research by other scholars. Understudied genes do not progress all the way through the pipeline, but it is unclear where this leak primarily occurs. The first possibility is that some genes are less studied because they are rarely identified as hits in experiments. Prior studies have, however, shown that understudied genes are frequent hits in high-throughput experiments (*Riba et al., 2016*; *Haynes et al., 2018*; *Stoeger and Nunes Amaral, 2020*), suggesting that this is not the case. The second possibility is that understudied genes are frequently found as hits in high-throughput experiments but are not investigated further by the authors. The final possibility is that subsequent studies do not continue work on understudied genes revealed by the initial study.

To evaluate these three possibilities, we gathered high-throughput experimental results and the titles and abstracts of articles reporting on these results, as well as the titles and abstracts of articles citing these reporting articles (focus studies). We then quantified the attention that 'hit' genes from these experiments had received by counting their occurrences in titles of abstracts within the entire biomedical literature excluding these high-throughput experiments. For example, the results for transcriptomics were obtained from 148 focus studies (original research) and 1678 subsequent studies (articles citing focus articles; green boxes *Figure 1B*). Those 148 studies identified 18,295 genes to be differentially expressed in at least one experiment but, in their titles and abstracts, mentioned only 161 of those 18,295 genes (*Figure 1B*). While the 18,295 genes have received similar research attention as the entirety of protein-coding genes (white box in *Figure 1B*), the 161 unique hit genes mentioned in the title or abstract are significantly better studied (p=1.4e-61 by two-side Mann-Whitney U test). The articles citing the focus articles also only mentioned in their titles and abstracts 692 unique genes that were differentially expressed in the cited focus study (irrespective of whether they were reported in title or abstract of the respective cited transcriptomic study). These 692 unique genes were similarly highly studied as those reported in the title and abstract of transcriptomic focus studies.

To demonstrate that these findings are not specific to any single type of high-throughput data, we considered CRISPR screens, transcriptomics, affinity purification-mass spectrometry, and GWAS. Evaluating the first possibility that some genes are less studied because they are rarely identified as hits in experiments, we found that understudied genes were frequently found as hits in high-throughput experiments (*Figure 1B*). This demonstrates, in line with earlier studies (*Riba et al., 2016*; *Haynes et al., 2018*; *Border et al., 2019*; *Stoeger and Nunes Amaral, 2020*; *Zhang et al., 2020*; *Byrne et al., 2022*), that the lack of publications on some genes is not explained by a lack of underlying biological experimental evidence.

We next evaluated the second possibility that understudied genes are frequently found as hits in high-throughput experiments but are not investigated further by the authors. We indeed found that hit genes that are highlighted in the title or abstract are over-represented among the 20% highest-studied genes in all biomedical literature (*Figure 1B*). These trends are independent of significance threshold (*Figure 1—figure supplement 5*) and (except for CRISPR screens) whether we considered the current scientific literature or literature published before 2003, before any of these articles had been published (*Figure 1—figure supplement 6*). We also find that this effect holds when controlling

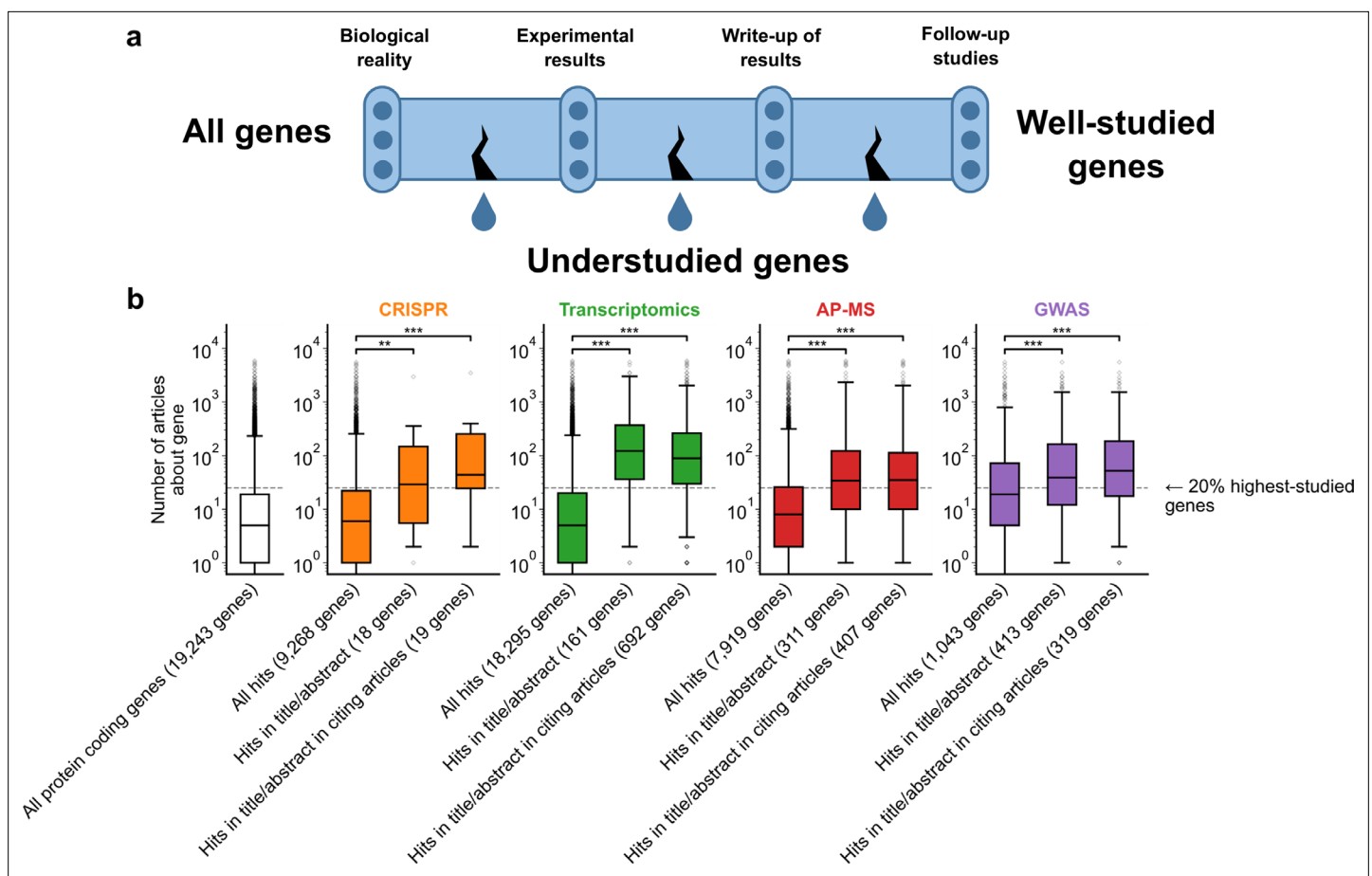

**Figure 1.** A shift in focus toward well-studied genes occurs during the summarization and write-up of results and remains in subsequent studies. (**a**) Conceptual diagram depicting possible points of abandonment for understudied genes in studies using high-throughput -omics experiments. (**b**) We identified articles reporting on genome-wide CRISPR screens (CRISPR, 15 focus articles and 18 citing articles), transcriptomics (T-omics, 148 focus articles and 1,678 citing articles), affinity purification–mass spectrometry (AP-MS, 296 focus articles and 1,320 citing articles), and GWAS (450 focus articles and 3,524 citing articles). Focusing only on protein-coding genes, we retrieved data uploaded to repositories describing which genes came up as 'hits' in each experiment. We then retrieved the hits mentioned in the titles and abstracts of those articles and hits mentioned in the titles and abstracts of articles citing those articles. Unique hit genes are only counted once. Bibliometric data reveals that understudied genes are frequently hits in -omics experiments but are not typically highlighted in the title/abstract of reporting articles, nor in the title/abstract or articles citing reporting articles. For example, the results for CRISPR were obtained from 15 focus studies (original research) and 18 subsequent studies (papers citing focus articles). Those 15 studies identified 9268 genes where loss-of-function changed phenotypes but, in their titles and abstracts, mentioned only 18 of those 9268 genes. While the 9268 hit genes have received similar research attention to the entirety of protein-coding genes, the 18 hit genes mentioned in the title or abstract are significantly better studied (p=0.0033 by two-side Mann-Whitney U test). The articles citing the focus articles also only mentioned in their titles and abstracts 19 highly-studied hit genes. ** denotes p<0.01 and *** denotes p<0.001 by two-sided Mann-Whitney U test, comparing genes highlighted in title/abstract to genes present in hit lists.

The online version of this article includes the following figure supplement(s) for figure 1:

**Figure supplement 1.** PRISMA diagram for the selection of genome-wide association studies (GWAS, from studies indexed by the NHGRI-EBI GWAS catalog [*Buniello et al., 2019*]).

**Figure supplement 2.** PRISMA diagram for the selection of affinity purification–mass spectrometry (AP-MS, indexed by BioGRID [*Oughtred et al., 2021*]).

**Figure supplement 3.** PRISMA diagram for the selection of transcriptomic studies (indexed by the EBI Gene Expression Atlas [*Papatheodorou et al., 2018*]).

**Figure supplement 4.** PRISMA diagram for the selection of genome-wide screens using CRISPR (indexed by BioGRID Open Repository of CRISPR Screens [*Oughtred et al., 2021*]).

**Figure supplement 5.** Variant of *Figure 1B* with alternate p-value/FDR thresholds for significance where applicable.

**Figure supplement 6.** Variant of *Figure 1B* only considering articles published in 2002 or before, prior to the publication of any of the articles featuring -omics experiments which we considered for this analysis.

*Figure 1 continued on next page*

**Figure supplement 7.** Variant of *Figure 1B* only considering one randomly chosen gene per article title/abstract.

for the number of genes in each title/abstract by only considering one randomly-chosen gene per title/abstract (*Figure 1—figure supplement 7*).

Understudied genes are least frequently highlighted in the title/abstract in transcriptomics experiments and most frequently highlighted in the title/abstract in CRISPR screens. GWAS studies tend to return better-studied genes as hits; the median hit gene in GWAS studies was more studied in the biomedical literature than 75% of genes. Hit median gene highlighted in the title/abstract in GWAS studies was more studied in the biomedical literature than 85% of all protein coding genes. This may explain the prior observation that the total number of articles on individual genes partially correlates with the total number of occurrences as a hit in GWAS studies (*Stoeger et al., 2018*).

Evaluating the final possibility, we found that the reception of -omics studies in later scientific literature either reproduced authors' initial selection of highly studied genes or slightly mitigated it. Jointly, the above findings reinforce that understudied genes become abandoned between the completion of -omics experiments and the reporting of results, rather than being abandoned by later research.

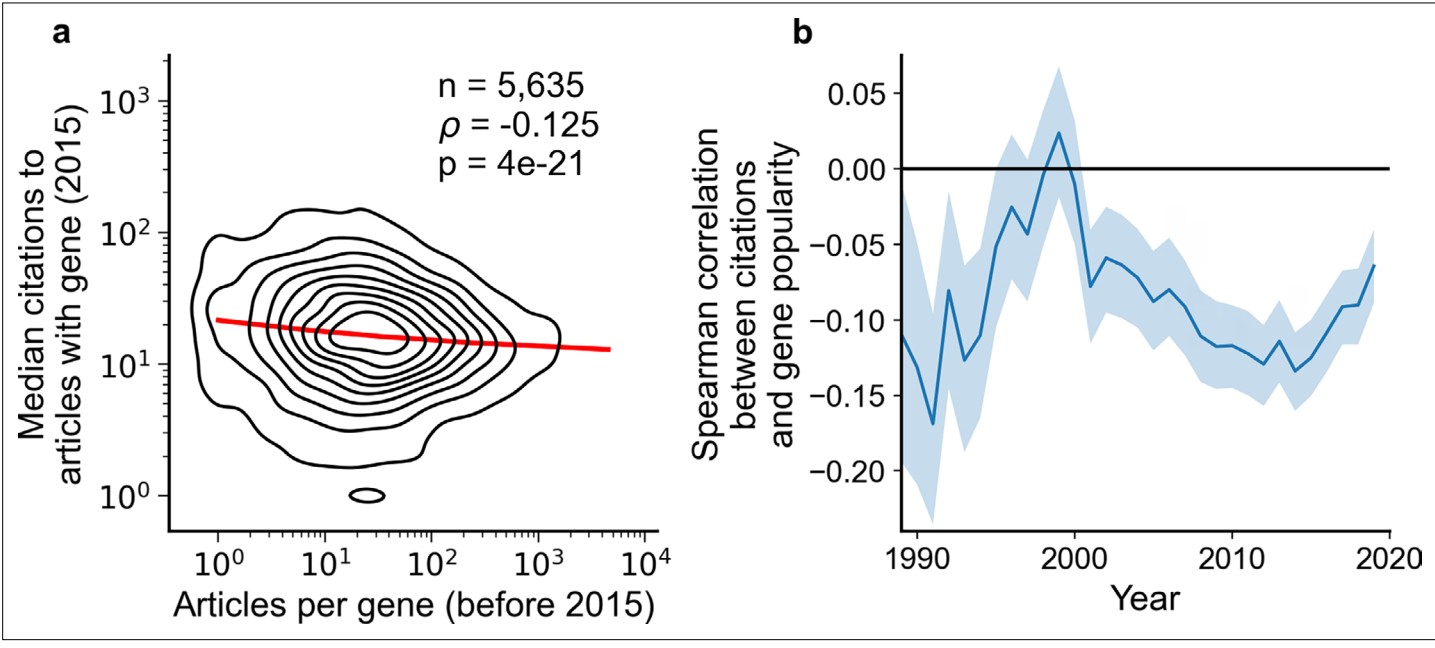

**Figure 2.** Articles focusing on less popular genes tend to accrue more citations. (**a**) Density plot shows correlation between articles per gene before 2015 and median citations to articles published in 2015. Contours correspond to deciles in density. Solid red line shows locally weighted scatterplot smoothing (LOWESS) regression. $\rho$ is Spearman rank correlation and p the significance values of the Spearman rank correlation as described by *Kendall and Stuart, 1973*. We forgo depicting more recent years than 2015 to allow for citations to accumulate over multiple years, providing a more sensitive and robust readout of long-term impact. (**b**) Spearman correlation of previous gene popularity (i.e. number of articles) to median citations per year since 1990. Solid blue line indicates nominal Spearman correlation, shaded region indicates bootstrapped 95% confidence interval (n=1000). Only articles with a single gene in the title/abstract are considered, excluding the 30.4% of gene-focused studies which feature more than one gene in the title/abstract. For more recent years, where articles have had less time to accumulate citations, insufficient signal may cause correlation to converge toward zero.

The online version of this article includes the following figure supplement(s) for figure 2:

**Figure supplement 1.** Likelihoods of being highly cited (top 5% of citations among all articles about genes), panel (**a**) or lowly cited (bottom 5% of citations among all articles about genes), panel (**b**) for articles about the most popular genes (top 5% accumulated articles) versus articles about the least popular genes (bottom 5% accumulated articles) by year of publication.

**Figure supplement 2.** Spearman correlation and significance (*Kendall and Stuart, 1973*; *Kendall and Stuart, 1961*) of normalized gene popularity vs normalized article citation rank for articles within disease MeSH terms from 2014 to 2018.

**Figure supplement 3.** Spearman correlation and significance (*Kendall and Stuart, 1973*; *Kendall and Stuart, 1961*) of normalized gene popularity vs normalized article citation rank for articles within technique-related MeSH terms from 2014 to 2018.

## Subsequent reception by other scientists does not penalize studies on understudied genes

The abandonment of understudied genes could be driven by the valid concern of biomedical researchers that focusing on less-investigated genes will yield articles with lower impact (*Kustatscher et al., 2022*), as observed around the turn of the millenium (*Pfeiffer and Hoffmann, 2007*). If this were the case, preemptively avoiding understudied hits would be the rational decision for authors of -omics studies.

We thus decided to complement our preceding analysis by an analysis explicitly focused on citation impact. Notably, we found that the concern of publications on understudied genes receiving fewer citations does not hold for present-day research on human genes; in biomedical literature at-large, articles focusing on less-investigated genes typically accumulate more citations, an effect that has held consistently since 2001 (*Figure 2*). Further, since 1990, articles about the least popular genes have at times been three to four times more likely to be among the most cited articles than articles on the most popular genes whereas articles on the most popular genes have been slightly less to be highly cited than lowly cited (*Figure 2—figure supplement 1*).

To rule out that these macroscopic observations stem from us having aggregated over different diseases, we separately analyzed 602 disease-related MeSH terms (*Figure 2—figure supplement 2*). We found 29 MeSH terms with a statistically significant Spearman correlation using Benjamini-Hochberg FDR <0.01 (*Supplementary file 1*), of which 27 showed a negative association and only 2 showed a positive association. This result again opposes the hypothesis that less-investigated genes will yield articles with lower impact.

Returning to our observation that understudied hits from high-throughput assays are not promoted to the title and abstract of the resulting publication, we next tested if different experimental approaches demonstrated distinct associations between gene popularity and citations (*Figure 2—figure supplement 3*). Among 264 technique-related MeSH terms tested, there were 20 MeSH terms with a statistically significant Spearman correlation using Benjamini-Hochberg FDR <0.01 (*Supplementary file 2*), of which 16 showed a negative association and only 4 showed a positive association. Notably, MeSH terms representing high-throughput techniques (e.g. D055106:Genome-Wide Association Study and D020869:Gene Expression Profiling) showed no significant association. This finding suggests that authors of high-throughput studies have little to gain or lose citation-wise by highlighting understudied genes.

To summarize, our investigations are reminiscent of the previously described separation between 'large-scale' and 'small-scale' biological research (*Knorr Cetina, 1999*; *Alberts, 1985*; *Richardson and Stevens, 2015*). Authors of high-throughput studies do not highlight understudied genes in the title or abstract of their publications, the sections of the publication most accessible to other scientists. While, overall, understudied genes (and high-throughput assays themselves [*Stoeger and Nunes Amaral, 2022*]) correlate with increased citation impact, for high-throughput studies any potential gain in citations is either absent or too small to be significant. Thus, there may not be any incentive for authors of high-throughput studies to highlight understudied genes.

## Identification of biological and experimental factors associated with selection of highlighted genes

To illuminate why understudied genes are abandoned between experimental results and the write-up of results, we performed a literature review to identify factors that have been proposed to limit studies of understudied genes (*Supplementary file 3*). These factors range from evolutionary factors (e.g. whether a gene only has homologs in primates), to chemical factors (e.g. gene length or hydrophobicity of protein product), to historical factors (e.g. whether a gene's sequence has previously been patented) to materialistic factors affecting experimental design (e.g. whether designed antibodies are robust for immunohistochemistry).

As any of these factors could plausibly affect gene selection within individual domains of biomedical research, we returned to the -omics data described above (*Figure 1B*) and measured how much these factors align with the selective highlighting of hit genes in the title or abstract of GWAS, AP-MS, transcriptomics, and CRISPR studies.

We identified 45 factors that relate to genes and found 33 (12 out of 23 binary factors and 21 out of 22 continuous factors) associated with selection in at least one assay type at Benjamini-Hochberg

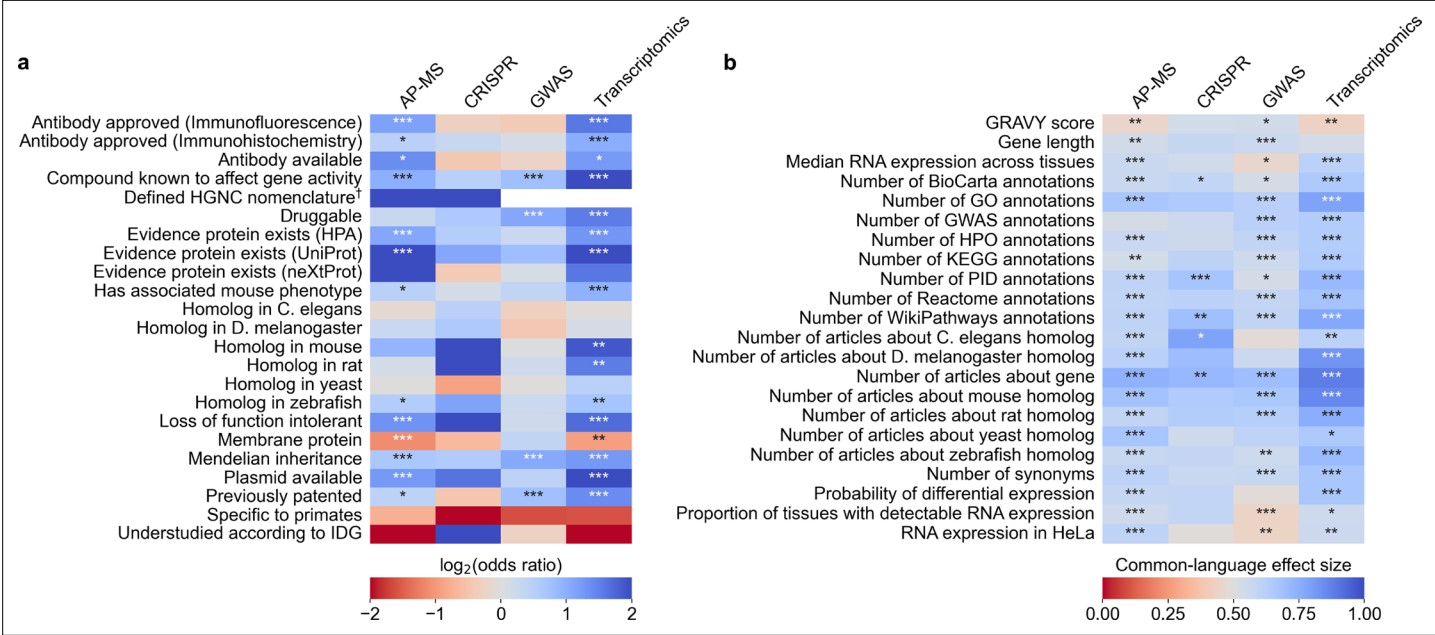

**Figure 3.** We evaluated which gene-related factors are associated with elevation to the title/abstract of an article featuring a high-throughput experiment. (**a**) Association between factors with binary (True/False) identities and highlighting hits in title/abstract of reporting articles. Values represent the odds ratio between hits in the collected articles and hits mentioned in the title or abstract of collected articles (e.g. hits with a compound known to affect gene activity are 4.262 times as likely to be mentioned in the title/abstract in an article using transcriptomics, corresponding to an odds ratio of 4.331). Collected articles are described in *Figure 1B* and *Figure 1—figure supplement 5*, *Figure 1—figure supplement 6* and *Figure 1—figure supplement 7*. 95% confidence interval of odds ratio is shown in parentheses. *=Benjamini-Hochberg FDR <0.05, **=FDR < 0.01, and ***=FDR < 0.001 by two-sided Fisher exact test. Results are shown numerically in *Supplementary file 4*. For consistency between studies, hits were restricted to protein-coding genes. Thus, status as a protein-coding gene could not be tested. †No genes without a defined HUGO symbol were found as hits in GWAS or transcriptomics studies. (**b**) Association with factors with continuous identities and highlighting hits in title/abstract of reporting articles. Values represent F, the common-language effect size (equivalent to AUROC, where ~0.5 indicates little effect, >0.5 indicates positive effect and <0.5 indicates negative effect) of being mentioned in the titles/abstracts of the collected articles described in *Figure 1B* and *Figure 1—figure supplement 5*, *Figure 1—figure supplement 6* and *Figure 1—figure supplement 7*. *=Benjamini-Hochberg FDR <0.05, **=FDR < 0.01, and ***=FDR < 0.001 by two-sided Mann-Whitney U test. Results are shown numerically in *Supplementary file 5*.

The online version of this article includes the following figure supplement(s) for figure 3:

**Figure supplement 1.** Identification of default factors in FMUG, manually chosen to represent different clusters of factors.

**Figure supplement 2.** Similarity matrix and clustermap showing Spearman correlation between factors across all human protein-coding genes.

FDR <0.001 (*Figure 3*, *Supplementary file 4* and *Supplementary file 5*). Across the four assay types, the most informative binary factor describes whether there is a plasmid available for a gene in the AddGene plasmid catalog. We cautiously hypothesize that this might reflect on many different research groups producing reagents surrounding the genes that they actively study. The most informative continuous factor is the number of research articles about a gene (*Figure 1B*).

To better understand how all 45 factors are related, we performed a cluster analysis of the collected factors (*Figure 3—figure supplement 1* and *Figure 3—figure supplement 2*). This clustering suggests that many factors influencing the abandonment of understudied genes are not independent. For instance, we find that the number of articles about a gene is heavily correlated with the number of annotations for that gene in all surveyed databases. In another case, gene length is heavily correlated with the number of GWAS annotations for a gene, as described before in terms of transcript length and single-nucleotide polymorphisms (*Lopes et al., 2021*).

## Data-driven design of a tool to promote the investigation of understudied genes

To promote the investigation of understudied genes, we combined all the above insights to create a tool we denoted *find my understudied genes* (FMUG). Our literature review revealed several tools and resources aiming to promote research of understudied genes by publicizing understudied genes

(*Duek et al., 2018*; *Crow et al., 2019*; *Perdigão and Rosa, 2019*; *Essegian et al., 2020*; *Sheils et al., 2021*; *Higgins et al., 2022*; *Wainberg et al., 2021*; *Rocha et al., 2023* or by providing information about hit genes *Stoeger et al., 2018*; *Rebhan et al., 1998*; *Tan et al., 2017*; *Kustatscher et al., 2019*; *Wu et al., 2021*; *Jiang et al., 2022*). However, we noted the absence of tools enabling scientists to actively engage with factors that align with gene selection. Although such factors are largely correlated when considering all genes (*Figure 3—figure supplement 1* and *Figure 3—figure supplement 2*), some factors cluster together and the influence of specific factors could vary across laboratories. For instance, scientists could vary in their ability to perform proteomics, or ability to explore orthologous genes in *C. elegans*, or ability to leverage human population data, or perform standardized mouse assays.

Our tool makes selection bias explicit, while acknowledging that different laboratories vary in their techniques and capabilities for follow-up research. Rather than telling scientists about the existence of biases, FMUG aims to prompt scientists to make bias-aware informed decisions to identify and potentially tackle important gaps in knowledge that they are well-suited to address. For this reason, we believe that FMUG will not be of value only to scientists engaging in high-throughput studies, but also by scientists wishing to mine existing datasets for hit genes that they would be well-positioned to investigate further.

FMUG takes a list of genes from the user (ostensibly a hit list from a high-throughput -omics experiment) and provides the kind of information that will allow a user to select genes for further study.

Users can employ filters that reflect the factors identified in our literature review and supported by our analysis. The default information provided to users consists of factors that are representative of the identified clusters (*Figure 3—figure supplement 1*) and strongly associated with gene selection in high-throughput experiments (*Figure 3*, *Supplementary file 4*, and *Supplementary file 5*). In extended options, users can select any factor that demonstrated a significant association with the selection of genes. For instance, a user may need to decide whether loss-of-function intolerant genes should be considered for further research or not, or whether there should be robust evidence that a gene is protein-coding. Some of these filters are context aware. For instance, a user may select genes that have already been studied in the general biomedical literature but not yet within the literature of their disease of interest.

To provide real-time feedback, users are, in parallel, presented the number of articles about genes in their initial input list and the number of articles about genes that passed their filters. Users can then export their filtered list of genes. In the interest of researcher privacy, FMUG keeps all information local to the user's machine. Usage of FMUG is illustrated in *Figure 4A* and demonstrated in *Video 1*. FMUG is developed in Flutter and is available for download at fmug.amaral.northwestern.edu as a MacOS/Windows app. For the development of custom software and analytical code, we provide the data underlying FMUG at https://github.com/amarallab/fmug_analysis, (copy archived at *Richardson and Stoenger, 2023*).

To determine the practical usefulness of FMUG to scientists, we used an early prototype of FMUG to identify understudied genes associated with aging. One of these genes was Splicing factor, proline- and glutamine-rich (*Sfpq*), which had not yet been investigated toward its role in biological aging. We found *Sfpq* to be transcriptionally downregulated during murine aging. Others had shown *Sfpq* to be required for the transcriptional elongation of long genes (*Takeuchi et al., 2018*). This led us to hypothesize that during vertebrate aging, the transcripts of long genes become downregulated in most tissues (*Figure 4B*). We found this hypothesis to be supported through a multi-species analysis which we published in December 2022 in Nature Aging (*Stoeger et al., 2022*), with another group publishing so in January 2023 in Nature Genetics (*Gyenis et al., 2023*), and a third group in iScience in March 2023 (*Ibañez-Solé et al., 2023*).

## Study limitations

Our study has several limitations. First, all analysis is subject to annotation errors in the various databases we employ. While these should be rare and not affect our overall findings, they may affect users who are interested in genes with discordant annotations. Second, we focus only on human genes. Different patterns of selection may exist for research on genes in other organisms. Third, we take a gene being mentioned in the title or abstract of an article as a proxy for a gene receiving attention

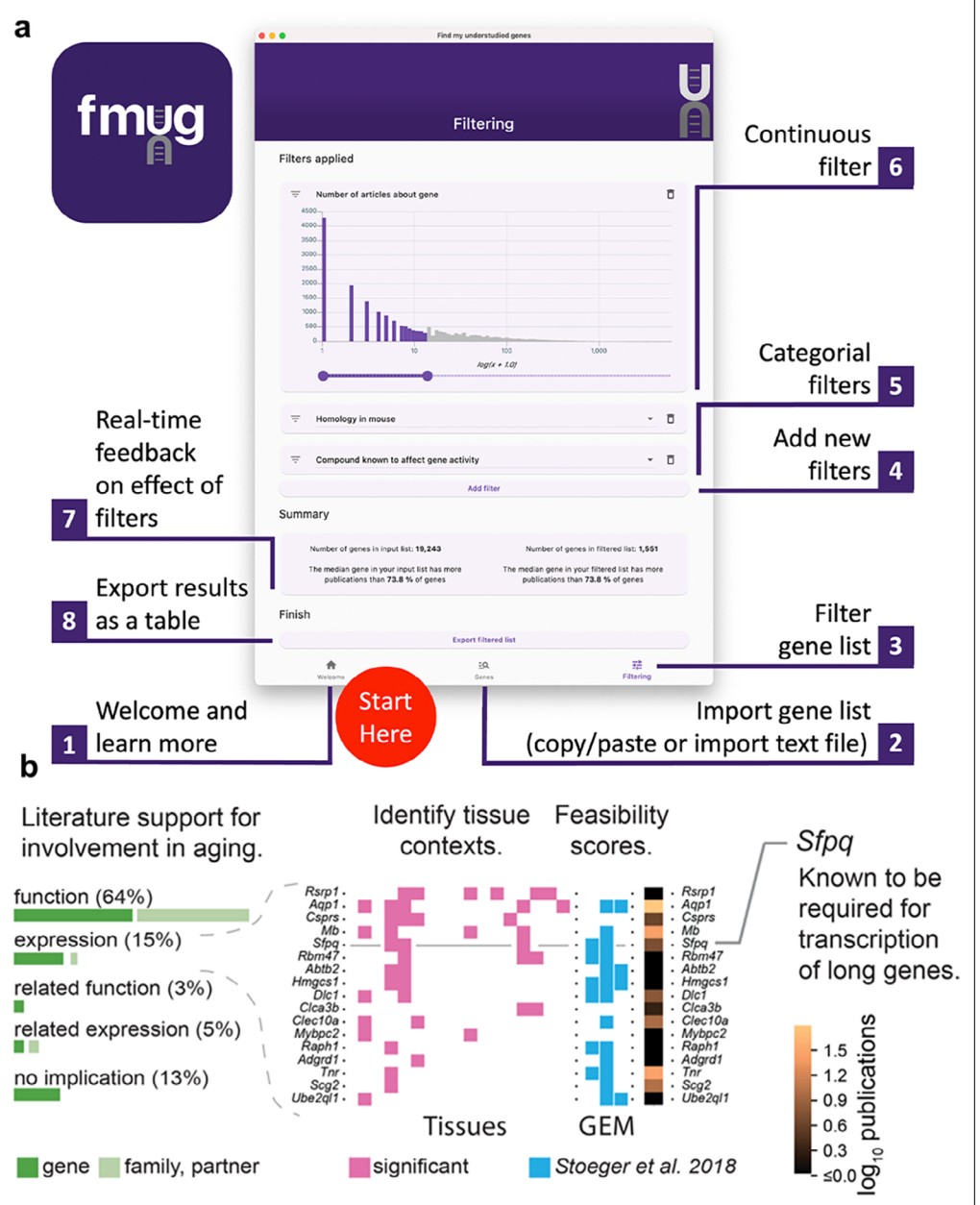

**Figure 4.** We created FMUG to help researchers identify understudied genes among their genes of interest and characterize their tractability for future research. (**a**) Diagram describing use of FMUG. (**b**) An early prototype of FMUG led us to the hypothesis that transcript length negatively correlates with up-regulation during aging. First, we identified genes that strongly associate with age-dependent transcriptional change across multiple cohorts. We then performed a literature review for each of these genes to identify the most direct way the genes (or evolutionarily closely related genes or functionally closely related partner proteins) had been studied in aging. Sixty-four percent had been functionally investigated in aging, 15% shown to change a measure of gene expression, 3% functionally investigated in a biological domain close to aging (such as senescence), and 5% shown to change a measure of gene expression in a biological domain close to aging. For genes reported by others to change expression with age, we identified tissues in which transcripts of the genes change during aging. We computed 'feasibility scores' scientific strategies (GEM: G: strong genetic support, E: and experimental potential, M: homolog in invertebrate model organism) as described by *Stoeger et al., 2018* and total number of publications in MEDLINE. Splicing factor, proline- and glutamine-rich (*Sfpq*) had previously been demonstrated by Takeuchi et al. to be required for the transcriptional elongation of long genes (*Takeuchi et al., 2018*). When performing a data-driven analysis of factors that could possibly explain age-dependent changes of the entire transcriptome, we thus included gene and transcript lengths, and subsequently found them to be more informative than transcription factors or microRNAs (*Stoeger et al., 2022*).

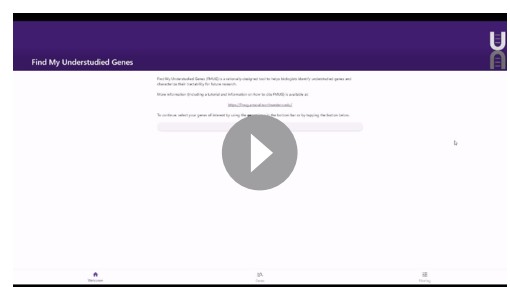

**Video 1.** FMUG tutorial video.
https://elifesciences.org/articles/93429/figures#video1

by the article's authors. The title and abstract are space-limited and thus cannot accommodate discussion of large numbers of genes.

Fourth, our literature review also identified further factors that we could not test more directly because of absent access to fitting data. These are: experts' tendency to deepen their expertise (*Edwards et al., 2011*), a perceived lack of accuracy of -omics studies (*Kustatscher et al., 2022*; *Brown and Peirson, 2018*), -omics serving research purposes beyond target gene identification (*Donohue and Love, 2024*), the absence of good protocols for mass spectrometry (*César-Razquin et al., 2015*), the electronic distribution and reading of research articles (*Evans, 2008*), rates of reproducibility (*Kustatscher et al., 2022*), career prospects of investigators (*Stoeger et al., 2018*; *Alberts et al., 2014*), authors beginning manuscripts with something familiar before introducing something new (*Uzzi et al., 2013*), and the human tendency to fall back to simplifying heuristics when making decisions under conditions with uncertainty (*Gilovich et al., 2002*). Fifth, we cannot resolve further which specific step between the conduct of an experiment and the writing of a research article leads to the abandonment of hit genes. Finally, we interpret the results of high-throughput experiments based on their representation in the NHGRI-EBI GWAS, BioGRID, EBI-GXA and BioGRID ORCS databases. The authors of the original studies may have processed their data differently, obtaining different results.

## Discussion

Efforts to address the gaps in detailed knowledge about most genes have crystallized as initiatives promoting the investigation of understudied sets of genes, an approach to gene scholarship recently termed 'unknomics' (*Rocha et al., 2023*). The insight that understudied genes are lost to titles and abstracts of research articles in a leaky pipeline between genome-wide assays and reporting of results, and FMUG, have already been useful in guiding our own unknomics research (*Figure 4B*).

As our present analysis is correlative, it also is tempting to propose controlled trials where published manuscripts on high-throughput studies randomly report hit genes in the abstract even if not investigated further by the authors. This intervention would need to be carefully designed since abstracts are limited in their size. Further, the observed discrepancy between the popularity of hits highlighted by GWAS versus other technologies suggests that some -omics technologies may be more powerful than others for characterizing understudied genes. This possibility merits further research and researchers participating in unknomics should consider the relative strengths of each technology toward providing tractable results for follow-up.

We believe that enabling scientists to consciously engage with bias in research target selection will enable more biomedical researchers to participate in unknomics, to the potential benefit of their own research impact and toward the advancement of our collective understanding of the entire human genome.

## Materials and methods

**Key resources table**

| Reagent type (species) or resource | Designation | Source or reference | Identifiers | Additional information |
|---|---|---|---|---|
| Software, algorithm | Python programming language | Python Software Foundation | RRID:SCR_008394 | |
| Software, algorithm | Find My Understudied Genes | This paper | RRID:SCR_025047 | Available at fmug.amaral.northwestern.edu |

## Genes information

*Homo sapiens* gene information was downloaded from NCBI Gene on Aug 16, 2022 [https://ftp.ncbi.nlm.nih.gov/gene/DATA/GENE_INFO/All_Data.gene_info.gz]. Only genes with an unambiguous mapping of Entrez ID to Ensembl ID were used (n=36,035). Number of gene synonyms, protein-coding status, and official gene symbol were derived from this dataset. A gene symbol was considered undefined if the gene's entry for HGNC gene symbol was '-'.

## Genes in title/abstract of primary research articles

*Homo sapiens* gene information was downloaded from NCBI Gene on Aug 16, 2022 [https://ftp.ncbi.nlm.nih.gov/gene/DATA/GENE_INFO/Mammalia/Homo_sapiens.gene_info.gz]. gene2pubmed was downloaded from NCBI Gene on August 16, 2022 [https://ftp.ncbi.nlm.nih.gov/gene/DATA/gene2pubmed.gz] (*Maglott et al., 2007*). PubTator gene annotations were downloaded from NIH-NLM on July 12, 2022 [https://ftp.ncbi.nlm.nih.gov/pub/lu/PubTatorCentral/] (*Maglott et al., 2007*; *Wei et al., 2019*). PubMed was downloaded on December 17, 2021 [https://ftp.ncbi.nlm.nih.gov/pubmed/baseline/].

Only using PMIDs annotated as primary research articles, a human gene was considered as mentioned in the title/abstract of the publication if gene was annotated as being in the title/abstract by PubTator and the article appeared in gene2pubmed.

## CRISPR articles

BioGRID ORCS (*Oughtred et al., 2021*) v1.1.6 was downloaded on April 25, 2022 [https://downloads.thebiogrid.org/BioGRID-ORCS/Release-Archive/BIOGRID-ORCS-1.1.6/]. Any genome-wide CRISPR knockout screens in human with an associated PubMed ID in which hit genes were mentioned in the title or abstract was considered (n=15). A total of 9268 unique genes were found as hits. Of these, 18 (0.19%) were highlighted in titles/abstracts in the reporting articles and 19 (0.21%) were highlighted in titles/abstracts in citing articles. A full list of PubMed IDs is available in *Supplementary file 6*.

## Transcriptomics articles

EBI-GXA (*Papatheodorou et al., 2018*) release 36 was downloaded on September 15, 2020 [https://web.archive.org/web/20201022184159/https://www.ebi.ac.uk/gxa/download]. This is the most recent release of EBI-GXA available as a bulk download. Any transcriptomics comparisons with an associated PubMed ID in which hit genes were mentioned in the title or abstract was considered (n=148). Analysis was restricted to protein-coding genes (some screens featured non-protein-coding genes, but this was not common to all analyses). DE was called at Benjamini-Hochberg FDR $q<0.05$. A total of 18,295 unique genes were found as hits. Of these, 161 (0.88%) were highlighted in titles/abstracts in the reporting articles and 692 (3.78%) were highlighted in titles/abstracts in citing articles. A full list of PubMed IDs is available in *Supplementary file 6*.

## Affinity purification–mass spectrometry articles

BioGRID (*Oughtred et al., 2021*) v3.5.186 was downloaded on April 25, 2022 [https://downloads.thebiogrid.org/BioGRID/Release-Archive/BIOGRID-3.5.186/]. Any interactions involving a human gene as the prey protein with an experimental evidence code of 'Affinity Capture-MS' labeled as 'High-Throughput' that had an associated PubMed ID in which hit genes were mentioned in the title or abstract was considered (n=296). Prey proteins in these interactions were considered hits. A total of 7919 unique genes were found as hits. Of these, 311 (3.93%) were highlighted in titles/abstracts in reporting articles and 407 (5.14%) were highlighted in titles/abstracts in citing articles. A full list of PubMed IDs is available in *Supplementary file 6*.

## GWAS articles

The NHGRI-EBI GWAS catalog (*Buniello et al., 2019*) (associations and studies) was download on August 17, 2022 [https://www.ebi.ac.uk/gwas/docs/file-downloads]. Any GWAS screens with an associated PubMed ID in which hit genes were mentioned in the title or abstract was considered (n=450). Only SNPs occurring within a gene were considered hits. A total of 1043 unique genes were found as hits. Of these, 413 (39.6%) were highlighted in titles/abstracts in reporting articles and 319 (30.6%)

were highlighted in titles/abstracts in citing articles. A full list of PubMed IDs is available in *Supplementary file 6*.

## Citing articles

NIH iCite v32 was downloaded on August 25, 2022 (*Hutchins and Santangelo, 2019*) [https://nih.figshare.com/collections/iCite_Database_Snapshots_NIH_Open_Citation_Collection_/4586573/32].

## Functional annotations

Mapping of genes to Gene Ontology / Protein Interaction Database / WikiPathways / Reactome / Kyoto Encyclopedia of Genes and Genomes / Human Phenotype Ontology / BioCarta categories was derived from MSigDB v7.5 Entrez ID.gmt files, downloaded on April 12, 2022 [http://www.gsea-msigdb.org/gsea/downloads_archive.jsp].

## Between-species homology

Homologene Build 68 was used to determine interspecies homology [https://ftp.ncbi.nih.gov/pub/HomoloGene/build68/]. Human = taxid:9606, mouse = taxid:10090, rat = taxid:10116, *c. elegans* = taxid:6239, *d. melanogaster* = taxid:7227, yeast = taxid:559292, zebrafish = taxid:7955.

## Primate specificity

Human genes were considered primate-specific if the only other members of their homology group belonged to primate genomes. Primate taxonomy ids were downloaded from NCBI Taxonomy on September 20, 2022 [https://www.ncbi.nlm.nih.gov/taxonomy/?term=txid9443[Subtree]].

## Number of publications in model organisms

Gene information was downloaded from NCBI Gene on August 16, 2022 [https://ftp.ncbi.nlm.nih.gov/gene/DATA/GENE_INFO/All_Data.gene_info.gz].

Only using PMIDs annotated as primary research articles, genes was considered as mentioned in the title/abstract of the publication if gene was annotated as being in the title/abstract by PubTator and the article appeared in gene2pubmed.

Genes in model organisms were mapped to human genes and the number of articles on those mapping to human genes were counted. If a model organism's gene had homology to human but no associated publications, the number of publications was resolved to zero. Otherwise, counts were listed as NA.

## Mouse phenotype hits

International Mouse Phenotyping Consortium data release 17.0 was downloaded on August 18, 2022 [https://www.mousephenotype.org/data/releasehttps://www.mousephenotype.org/data/release]. Mouse genes were matched to human genes with Homologene.

## Gene expression atlas (EBI-GXA)

EBI-GXA release 36 was downloaded on September 15, 2020 [https://web.archive.org/web/20201022184159/https://www.ebi.ac.uk/gxa/download]. This is the most recent release of EBI-GXA available as a bulk download. For probability of DE, only RNA-seq comparisons were considered and DE was called at Benjamini-Hochberg $q<0.05$.

## Global RNA expression

RNA consensus tissue gene data from HPA release 21.1 was downloaded on September 20, 2022 [https://www.proteinatlas.org/about/download]. Global RNA expression was estimated by taking the median expression (nTPM) across tissues for each gene and the proportion of tissues with detectable (≥1 nTPM) expression for each gene.

## Expression in HeLa cells

RNA cell line gene data from HPA release 21.1 was downloaded on September 20, 2022 [https://www.proteinatlas.org/about/download]. Expression is in nTPM.

## Previous patent activity

Genes with patent activity were defined from Table S1 of *Rosenfeld and Mason, 2013*. Genes were mapped with their HGNC symbol. This analysis aligned sequences in patents to the human genome to estimate patent coverage of human coding sequences. Although this does not necessarily reflect whether the mapped genes were claimed directly by the patent holder, as noted by others (*Tu et al., 2014*), this analysis remains the most comprehensive available for determining patent coverage of the human genome.

## Druggability

Druggable genes were identified from Table S1 of *Finan et al., 2017*. Genes were mapped with their Ensembl identifier.

## Gene length

GenBank was downloaded in spring 2017 (genome version GRCh38.p10). Gene length is defined here as the span of the longest transcript on the chromosome. This aligns with the model of gene length used in *Stoeger et al., 2018*.

## Solubility

SwissProt protein sequences and mapping tables to Entrez GeneIDs were downloaded from Uniprot in spring 2017. Protein GRAVY score (ignoring Pyrrolysine and Selenocysteine) was estimated with BioPython (*Cock et al., 2009*).

## Loss-of-function intolerance

Data was obtained from *Karczewski et al., 2020*. pLI scores >0.9 on main transcripts, as flagged by authors, were considered as highly loss-of-function intolerant as described by *Lek et al., 2016*.

## Number of GWAS hits

EBI GWAS catalog (*Buniello et al., 2019*; associations and studies) was download on August 17, 2022 [https://www.ebi.ac.uk/gwas/docs/file-downloads]. Loci were mapped to the nearest gene.

## Status as understudied protein

The Illuminating the Druggable Genome understudied protein list was downloaded on September 20, 2022 [https://github.com/druggablegenome/IDGTargets/blob/master/IDG_TargetList_Current-Version.json].

## Human protein atlas

HPA release 21.1 was downloaded on September 20, 2022 [https://www.proteinatlas.org/search]. Evidence for a protein's existence, as determined by NeXtProt, HPA, or UniProt was resolved as True if the respective evidence entry was annotated as 'Evidence at protein level'. Status as a membrane protein was determined by whether the 'Protein class' column contained the string 'membrane protein'. Antibodies were considered available for each protein if the protein's entry in the 'Antibody' column was not null.

## Availability of plasmids

The AddGene plasmid catalog was downloaded on August 12, 2022 [https://www.addgene.org/browse/gene/gene-list-data/?_=1666368044314].

## Availability of compounds

The catalog of gene targets was downloaded from ChEMBL on September 20, 2022 [https://www.ebi.ac.uk/chembl/g/#browse/targets]. UniProt IDs were converted to Entrez IDs to identify which human genes were affected by any compound.

## Mendelian inheritance

Autosomal dominant [https://hpo.jax.org/app/browse/term/HP:0000006] and autosomal recessive [https://hpo.jax.org/app/browse/term/HP:0000007] inherited disease-gene associations were downloaded from the Human Phenotype Ontology on September 20, 2022. Genes were considered to have evidence of Mendelian inheritance if they appeared in these lists of associations.

## Acknowledgements

We thank Xiaojing Sui for testing FMUG and Northwestern Information Technology for technical assistance. RAKR was supported in part by the National Institutes of Health Training Grant (T32GM008449) through Northwestern University's Biotechnology Training Program. RAKR also acknowledges support from the Dr. John N Nicholson fellowship from Northwestern University and Moderna Inc, "Identifying bias and improving reproducibility in RNA-seq computational pipelines". LANA was supported by NSF 1956338, NIH U19AI135964 and Simons Foundation DMS-1764421. TS was supported by NIH K99AG068544 and NIH KR00AG068544. We thank Alexander Misharin, Richard Morimoto, and Scott Budinger for feedback on an early prototype of FMUG which we used as part of our shared research into the biology of aging.

## Additional information

### Funding

| Funder | Grant reference number | Author |
| --- | --- | --- |
| National Institute on Aging | R00AG068544 | Thomas Stoeger |
| National Institute on Aging | K99AG068544 | Thomas Stoeger |
| National Institute of Allergy and Infectious Diseases | U19AI135964 | Luis A Nunes Amaral |
| National Institute of General Medical Sciences | T32GM008449 | Reese Richardson |
| National Science Foundation | 1956338 | Luis A Nunes Amaral |
| Simons Foundation | DMS-1764421 | Luis A Nunes Amaral |

The funders had no role in study design, data collection and interpretation, or the decision to submit the work for publication.

### Author contributions

Reese Richardson, Conceptualization, Resources, Data curation, Software, Formal analysis, Funding acquisition, Methodology, Writing – original draft, Writing – review and editing; Heliodoro Tejedor Navarro, Software; Luis A Nunes Amaral, Conceptualization, Supervision, Visualization, Project administration, Writing – review and editing; Thomas Stoeger, Conceptualization, Resources, Supervision, Project administration, Writing – review and editing

### Author ORCIDs

Reese Richardson ⓘ https://orcid.org/0000-0002-6058-5886
Luis A Nunes Amaral ⓘ https://orcid.org/0000-0002-3762-789X
Thomas Stoeger ⓘ https://orcid.org/0000-0002-5540-4278

Reviewer #1 (Public Review): https://doi.org/10.7554/eLife.93429.3.sa1
Reviewer #2 (Public Review): https://doi.org/10.7554/eLife.93429.3.sa2
Reviewer #3 (Public Review): https://doi.org/10.7554/eLife.93429.3.sa3
Author Response https://doi.org/10.7554/eLife.93429.3.sa4

## Additional files

### Supplementary files
• Supplementary file 1. Disease-related MeSH terms with a significant association between gene popularity and citations.
• Supplementary file 2. Technique-related MeSH terms with a significant association between gene popularity and citations.
• Supplementary file 3. Collected factors from literature search.
• Supplementary file 4. Association between factors with binary (True/False) identities and highlighting hits in title/abstract of reporting articles.
• Supplementary file 5. Association with factors with continuous identities and highlighting hits in title/abstract of reporting articles.
• Supplementary file 6. PubMed ID's and mentioned genes for collected GWAS, CRISPR, transcriptomics and AP-MS articles.
• MDAR checklist

### Data availability
All underlying data for figures are available on GitHub, (copy archived at *Richardson and Stoenger, 2023*). Code for the FMUG app is also available on GitHub. Details on existing datasets used in this study are available in Materials and methods.

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
