## [Editor Report · eLife assessment]

This study investigated the factors related to understudied genes in biomedical research. It showed that understudied genes are largely abandoned at the writing stage, and it identified a number of biological and experimental factors that influence which genes are selected for investigation. The study is an **important** contribution to this branch of meta-research, and the evidence in support of the findings is **solid**.

---

## [Referee Report · Reviewer #1 (Public Review)]

The authors have addressed most of the concerns I had about the original version in this revised version.

---

## [Referee Report · Reviewer #3 (Public Review)]

The message conveyed by figure 1b is now clearer, but could still be improved. The authors explained the meaning of this figure well in their response to the reviewers: "For example, the results for CRISPR were obtained from 15 focus studies (original research) and 18 subsequent studies (papers citing focus articles). Those 15 studies identified 9,268 genes where loss-of-function changed phenotypes but, in their titles and abstracts, mentioned only 18 of those 9,268 genes. While the 9,268 hit genes have received similar research attention to the entirety of protein-coding genes, the 18 hit genes mentioned in the title or abstract are significantly more well studied. The articles citing the focus articles also only mentioned in their titles and abstracts 19 highly studied hit genes".

The new Figure S8 is good.

---

## [Author Response]

The following is the authors’ response to the original reviews.

**eLife Assessment**
This study investigated the factors related to understudied genes in biomedical research. It showed that understudied genes are largely abandoned at the writing stage, and it identified a number of biological and experimental factors that influence which genes are selected for investigation. The study is a valuable contribution to this branch of meta-research, and while the evidence in support of the findings is solid, the interpretation and presentation of the results (especially the figures) needs to be improved.

We thank the editor and reviewers for their detailed and thoughtful assessment of our work. Below, we present detailed responses to reviewers’ comments and suggestions. We are also submitting a version edited for clarity of presentation and precision of interpretation.

Following the eLife assessment, we also tried to identify further statements where results could be presented in a more precise way.

First, in the section Subsequent reception by other scientists does not penalize studies on understudied genes, we now state “This result again opposes the hypothesis that less-investigated genes will yield articles with lower impact.”

Second, in section Identification of biological and experimental factors associated with selection of highlighted genes, we now state:

“We cautiously hypothesize that this might reflect on many different research groups producing reagents surrounding the genes that they actively study. The most informative continuous factor is the number of research articles about a gene (Figure 1B).”, removing claims of causality.

Finally, for improved readability, we have moved all supplemental tables into separate .xlsx files.

**Reviewer #1 (Public Review):**
Summary and strengthsThe authors tried to address why only a subset of genes are highlighted in many publications. Is it because these highlighted genes are more important than others? Or is it because there are non-genetic reasons? This is a critical question because in the effort to discover new genes for drug targets and clinical benefit, we need to expand a pool of genes for deep analyses. So I appreciate the authors' efforts in this study, as it is timely and important. They also provided a framework called FMUG (short for Find My Understudied Gene) to evaluate genes for a number of features for subsequent analyses.

We thank the reviewer for their insightful comments and are pleased that the reviewer shares our appreciation for the gravity of these questions. As the reviewer emphasizes, it is critical to understand whether the choice of genes reflects their importance or non-genetic reasons. Previously we and others demonstrated that this choice does not reflect biological importance, when the latter is assessed through unbiased genome-wide data (e.g.: Haynes et al., 2018; Stoeger et al. 2018). Now we contribute to this critical question by systematically evaluating individual non-genetic reasons. We address the reviewer’s comments below.

WeaknessesMany of the figures are hard to comprehend, and the figure legends do not sufficiently explain them.For example, what was plotted in Fig 1b? The number of articles increased from results -> write-ups -> follow-ups in all four categories with different degrees. But it does not seem to match what the authors meant to deliver.

We apologize for the lack of clarity. We identified two interrelated elements that we have now fixed: (i) the prior figure legend provided for each genomics approach n number of articles, such as “GWAS (n=450 articles)”; (ii) the prior y-axis was labelled “Number of articles”.

Addressing the first element, we now rephrased the legend for clarity:

“b, We identified articles reporting on genome-wide CRISPR screens (CRISPR, 15 focus articles and 18 citing articles), transcriptomics (T-omics, 148 focus articles and 1,678 citing articles), affinity purification–mass spectrometry (AP-MS, 296 focus articles and 1,320 citing articles), and GWAS (450 focus articles and 3,524 citing articles). Focusing only on protein-coding genes (white box plot), we retrieved data uploaded to repositories describing which genes came up as “hits” in each experiment (first colored box plot). We then retrieved the hits mentioned in the titles and abstracts of those articles (second colored box plot) and hits mentioned in the titles and abstracts of articles citing those articles (third colored box plot). Unique hit genes are only counted once.”

The number of genes in each box plot is now reported in the x-axis labels for each step. For example, the results for CRISPR were obtained from 15 focus studies (original research) and 18 subsequent studies (papers citing focus articles). Those 15 studies identified 9,268 genes where loss-of-function changed phenotypes but, in their titles and abstracts, mentioned only 18 of those 9,268 genes. While the 9,268 hit genes have received similar research attention to the entirety of protein-coding genes, the 18 hit genes mentioned in the title or abstract are significantly more well studied. The articles citing the focus articles also only mentioned in their titles and abstracts 19 highly studied hit genes.

Addressing the second element, we updated the axis label to “Number of articles about gene”, to distinguish it from number of articles mentioned in the legend, convey that this is the number of articles about each gene that were published independently of the genomics assays we inspect. To further underscore this point we now label the “20% highest-studied genes” that we mention in the main text, and reworded the figure caption to better capture where the critical increase occurs: “A shift in focus towards well-studied genes occurs during the summarization and write-up of results and remains in subsequent studies.”.

Fig 4 is also confusing. It appears that the genes were clustered by many features that the authors developed. But does it have any relationship with genes being under- or over-studied?

We again apologize for the lack of clarity. As is described in the main text, while the results of Figs. 1-2 suggest that gene popularity may be predict the highlighting of a differentially expressed gene in the title or abstract, we want to conduct a systematically analysis of the factors that correlate with such a decision. We thus build a set of 45 factors that have been discussed as factors explaining why some genes receive increased research attention.

The data in Fig. 4 shows that those 45 factors are not independent but that some are highly correlated. Because of those correlations, we are able to select a smaller number as representative of the full set. Those are the default factors shown to users of FMUG. While users can choose all factors that are significantly correlated with the highlighting in title or abstract, the default of presenting factors representing different clusters of factors enabled us to limit the number of factors that are initially displayed.

Please note that following the suggestion of Reviewer 3, we have now moved this Figure to the supplemental material, as Figure S11.

**Reviewer #2 (Public Review)**
Summary and strengthsIn this manuscript the authors analyse the trajectory of understudied genes (UGs) from experiment to publication and study the reasons for why UGs remain underrepresented in the scientific literature. They show that UGs are not underrepresented in experimental datasets, but in the titles and abstracts of the manuscripts reporting experimental data as well as subsequent studies referring to those large-scale studies. They also develop an app that allows researchers to find UGs and their annotation state. Overall, this is a timely article that makes an important contribution to the field. It could help to boost the future investigation of understudied genes, a fundamental challenge in the life sciences. It is concise and overall well-written, and I very much enjoyed reading it. However, there are a few points that I think the authors should address.

We thank the reviewer for their kind assessment.

WeaknessesThe authors conclude that many UGs "are lost" from genome-wide assay at the manuscript writing stage. If I understand correctly, this is based on gene names not being reported in the title or abstract of these manuscripts. However, for genome-wide experiments, it would be quite difficult for authors to mention large numbers of understudied genes in the abstract. In contrast, one might highlight the expected behaviour of a well-studied protein simply to highlight that the genome-wide study provides credible results.

We agree that it is not reasonable to expect a title or abstract to highlight hundreds or even thousands of differentially expressed genes. We’ve now extended our Study Limitations section to address this:

“we take a gene being mentioned in the title or abstract of an article as a proxy for a gene receiving attention by the article’s authors. The title and abstract are space-limited and thus cannot accommodate discussion of large numbers of genes.”

We also agree that highlighting the expected behavior of a well-studied protein may provide credibility to a study and increase confidence on other results. The soundness of such a strategy was quantitatively studied in a study by Uzzi et al. (Science 2013), which we now include in the section on study limitations as:

“authors beginning manuscripts with something familiar before introducing something new”.

To convey the practical limitation of abstracts needing to be concise, we added the following sentence to our discussion section, when suggesting controlled trials that add genes to abstracts:

“This intervention would need to be carefully designed since abstracts are limited in their size.”

To avoid over-interpretation we have in the discussion also extended the sentence on “lost in a leaky pipeline” to “lost to titles and abstracts of research articles in a leaky pipeline”.

Our focus on titles and abstracts has been equally motivated by their availability (full text still is often behind paywalls and/or not accessible for bulk-download and text-mining) and by abstracts being the most visible and most read parts of research articles (e.g.: bioRxiv estimates that for the preprint for the present manuscript, the abstract was read ~10 times more frequently than full-text HTML and 4 times more frequently than the pdf).

Could this bias the authors' conclusions and, if so, how could this be addressed? For example, would it be worth to normalise studies based on the total number of genes they cover?

We previously described that – in line with the reviewer’s expectations – unstudied genes are preferentially added to the title or abstract of articles that feature more genes in the title or abstract (Stoeger et al., Plos Biology, 2022; Fig. 2B). Normalizing by the total number of genes should thus preserve the pronounced division between well-studied genes and unstudied genes show in Figure 1B. In line with these predictions, we randomly select one gene per title/abstract and find that the effect remains (see new Figure S7).

**Author response image 1. sa4fig1:** 

Figure 1B is confusing in its present form. I think the plot and/or the legend need revising. For example, what "numbers to the right of each box plot" are the authors referring to? Also, I assume that the filled boxes are understudied genes and the empty/white box is "all genes", but that's not explained in the legend. In the main text, the figure is referred to with the sentence "we found that hit genes that are highlighted in the title or abstract are strongly over-represented among the 20% highest-studied genes in all biomedical literature". I cannot follow how the figure shows this. My interpretation is that the y-axis is not showing the number of articles, but represents the percentage of articles mentioning a gene in the title/abstract, displayed on a log scale. If so, perhaps a better axis labels and legend text could be sufficient. But then one would also need to somehow connect this to the statement in the main text about the 20% highest-studied genes (a dashed line?). Alternatively, the authors could consider other ways of plotting these data, e.g. simply plotting the "% of publication in which a gene appears" from 0-100% or so.

Reviewer 1 raised a similar point on overall figure clarity. We identified two interrelated elements that contribute to overall confusion and have now fixed them (see response to Reviewer 1 beginning on page 2 of this document).

We attempted an alternative plotting of Fig 1B according to the reviewer’s suggestion. In the version below, the y-axis instead shows the percent of gene-related articles that are about each gene. We chose to keep the original y-axis (showing number of articles about each gene) as it additionally conveys the absolute scale of scholarship on individual genes.

**Author response image 2. sa4fig2:** 

**Reviewer #3 (Public Review):**
Summary and strengthsThe manuscript investigated the factors related to understudied genes in biomedical research. It showed that understudied are largely abandoned at the writing stage and identified biological and experimental factors associated with selection of highlighted genes.It is very important for the research community to recognize the systematic bias in research of human genes and take precautions when designing experiments and interpreting results. The authors have tried to profile this issue comprehensively and promoted more awareness and investigation of understudied genes.

We thank the reviewer for their kind assessment of our work.

WeaknessesRegarding result section 1 "Understudied genes are abandoned at synthesis/writing stage", the figures are not clear and do not convey the messages written in the main text. For example, in Figure 1B, figure S5 and S6,There is no "numbers to the right of each box plot".

The “numbers to the right” statement in the caption was an erroneous inclusion from an earlier version of the figure. We apologize for our error and have now removed this statement.

Do these box plots only show understudied genes? How many genes are there in each box plot? The definition and numbers of understudied genes are not clear.

The x-axis describes genes featured in each stage of the publication process (from all protein-coding genes to genes found as hits in genome-wide screen to genes found in the title/abstract to genes found in the title/abstract of citing articles) and the y-axis describes the number of articles annotated to those genes. We have also now added the number of genes in each box plot to the figure. This information is also in Materials and Methods under each technology’s heading (see also response to Reviewer 1 beginning on page 2 of this document).

**Author response image 3. sa4fig3:** 

"We found that hit genes that are highlighted in the title or abstract are strongly over-represented among the 20% highest-studied genes in all biomedical literature (Figure 1B)". This is not clear from the figure.

We have revised Figure 1B and its caption to better communicate the main point of the figure: that genes which make it to the title/abstract of the reporting article tend to be more popular than genes which are hits in genome-wide experiments from those articles. We have added a horizontal line that shows the cutoff for the top 20% most popular genes.

Regarding result section 2 "Subsequent reception by other scientists does not penalize studies on understudied genes", the authors showed in figure 2 that there is a negative correlation between articles per gene before 2015 and median citations to articles published in 2015. Another explanation could be that for popular genes, there are more low-quality articles that didn't get citations, not necessarily that less popular genes attract more citations.

We believe that both explanations for the observed phenomenon are not mutually exclusive. Previously, we focused on the median of citations to articles about a gene to capture the typical effect. In a new analysis, we also find support for the possibility outlined by the reviewer and believe that adding this to our manuscript complements and balances our analysis of citations. Specifically, in the new Figure S8B we find that most popular genes are slightly more likely to be among least cited papers (and in Figure S8A that the least studied genes have been much more likely to be among the most cited papers). In-text, we state:

“Further, since 1990, articles about the least popular genes have at times been 3 to 4 times more likely to be among the most cited articles than articles on the most popular genes whereas articles on the most popular genes have been slightly less to be highly cited than lowly cited (Figure S8)”.

We thank the reviewer for their suggestion, which strengthens our manuscript. The figure caption reads:

“Figure S8: Likelihoods of being highly cited (top 5% of citations among all articles about genes, panel a) or lowly cited (bottom 5% of citations among all articles about genes, panel b) for articles about the most popular genes (top 5% accumulated articles) versus articles about the least popular genes (bottom 5% accumulated articles) by year of publication. Only articles with a single gene in the title/abstract are considered. Shaded regions show ±1 standard error of the proportion."

**Author response image 4. sa4fig4:** 

Regarding result section 3 "Identification of biological and experimental factors associated with selection of highlighted genes", in Figure 3 and table s2, the author stated that "hits with a compound known to affect gene activity are 5.114 times as likely to be mentioned in the title/abstract in an article using transcriptomics", The number 5.144 comes out of nowhere both in the figure and the table. In addition, figure 4 is not informative enough to be included as a main figure.

This is the result of both a typo and imprecise terminology. The number should read 4.262 (the likelihood ratio of being mentioned in the title/abstract between genes with and without a compound), which corresponds to an odds ratio of 4.331. We have clarified this in the table caption, stating:

“e.g. hits with a compound known to affect gene activity are 4.262 times as likely to be mentioned in the title/abstract in an article using transcriptomics, corresponding to an odds ratio of 4.331".

We have removed Figure 4 as a main-text figure and added a version, with revised color scheme along comments of Reviewer 1, as Figure S11. We added to the figure caption “Bold indicates FMUG ‘s default factors, which we selected based on this clustering and based on their strength of association with gene selection (Figure 3, Table S2 and Table S3)."

**Recommendations for the authors:**

**Reviewer #1 (Recommendations for the authors):**
Fig 2a shows that papers highlighting understudied genes are actually cited more. I wonder why authors only looked at data before 2015. Fig 2b shows an increased correlation since 2015. Please consider redrawing Fig 2a to include data from 2015-2020?

We highlight data from 2015 since, from our used version of iCite (v32, released July 2022, covering citations made through most of 2021), papers published in 2015 have had about 6 years to accumulate citations. With fewer years to accumulate citations, insufficient signal may cause correlation to converge toward zero. Below, we repeat the analysis in Figure 2 but only considering citations made within a year of an article’s publication, which substantially reduces correlation (although remaining significant).

**Author response image 5. sa4fig5:** 

We added a note to the figure caption:

“We forgo depicting more recent years than 2015 to allow for citations to accumulate over multiple years, providing a more sensitive and robust readout of long-term impact.”

For Figure 2B, we add:

“For more recent years, where articles have had less time to accumulate citations, insufficient signal may cause correlation to converge toward zero.”

Can FMUG be posted on the web for easy access by researchers with non-computational backgrounds?"

We presently regretfully do not have the resources to create or maintain a web-based version. We hope that the publication of this manuscript will enable us to attract resources to create and maintain a web-based version.

**Reviewer #2 (Recommendations for the authors):**
Related to the first weakness in my public review: The observed disparity between CRISPR and GWAS study in terms of which genes they promote to the abstract is interesting. I wonder if this has to do with the application of these techniques. GWAS studies will often highlight that they retrieve known associations between a gene and a phenotype, to show that a screen is working. I guess often the point is to subsequently identify more genes associated with a particular phenotype, but often it is unclear how to validate/verify newly found associations. In contrast, CRISPR screens might be more focussed on functionally/mechanistically understanding unknown processes, e.g. observing a phenotype that appears/disappears in response to a gene deletion. In such studies, the follow-up of a previously unknown gene could be more straightforward and relevant to the outcome. Does that mean CRIPSR screens are better than GWAS studies for addressing the UG problem? Perhaps the authors could briefly discuss this issue.

The number of studies we included featuring CRISPR screens is relatively small (n = 15 compared to n = 450 for GWAS). Thus, it is not possible to conclude in a statistically sound manner whether authors of CRISPR screens are truly more likely to highlight understudied genes.

However, the reviewer raises compelling reasons for why this might be the case, and we now embed the broader discussion point that some techniques might be more powerful toward understudied genes.

The discussion now includes:

“Further, the observed discrepancy between the popularity of hits highlighted by GWAS versus other technologies suggests that some -omics technologies may be more powerful than others for characterizing understudied genes. This possibility merits further research and researchers participating in unknomics should consider the relative strengths of each technology towards providing tractable results for follow-up.”

Affinity capture mass spectrometry (Aff-MS): Perhaps I misunderstood this, but typically this is referred to as affinity purification MS (AP-MS)

Thank you for the clarification. We have changed ‘Aff-MS’ to ‘AP-MS’ throughout the manuscript.

Page 3, line 96. The sentence "The first possibility is that seemingly understudied genes are, in fact, not understudied as they would rarely be identified through experiments.". Would they not still be understudied, just not intentionally?

We have rephrased this sentence to:

“The first possibility is that some genes are less studied because they are rarely identified as hits in experiments.”

Fig 4 is very interesting, but I also found it a bit confusing. First, the choice of colour scheme, where blue shows the absence and white shows the presence of something, seems counterintuitive, especially on a white background. Second, I find it confusing that only some of the experiments are labelled in the heatmap. Could the authors not simply use Fig S9 as Fig 4? Or alternatively, only include the 8 labelled factors in the simplified figure.

In line with this feedback and that of Review #1 and #3, we have removed Figure 4 as a main-text figure and instead include this figure as Supplementary Figure S11. We have reversed the color scheme so that purple indicates one and white indicates zero. We also now label all factors. Previously we had only listed the default features of FMUG. We also now updated the figure legend to convey how it assisted the choice of default factors in FMUG. It reads:

“Bold indicates FMUG ‘s default factors, which we selected based on this clustering and based on their strength of association with gene selection (Figure 3, Table S2 and Table S3)”.

The FMUG app is fantastic and sounds exactly like something that is required to boost the visibility of understudied genes and overcome the understudied gene bias. However, I did not understand the choice of reporting this in the Discussion section.

We thank the reviewer for their enthusiasm, and have now moved FMUG into the results section.

To further increase usability of the FMUG app, is there a way it could be deployed online? I appreciate this could require a major amount of coding work, which would not be reasonable to demand. So please consider this a suggestion, potentially for a future implementation.

We presently regretfully do not have the resources to create or maintain a web-based version. We hope that the publication of this manuscript will enable us to attract resources to create and maintain a web-based version.

**Reviewer #3 (Recommendations for the authors):**
Table s2 and s3: p values are indicated by star signs. However, with so many hypothesis tests, the p values should be corrected for multiple tests.

We have now applied Benjamini-Hochberg multiple hypothesis correction to these tables, correcting p-values within each of the four technologies. We update our significance calling to read:

“We identified 45 factors that relate to genes and found 33 (12 out of 23 binary factors and 21 out of 22 continuous factors) associated with selection in at least one assay type at Benjamini-Hochberg FDR < 0.001.”

Figure S1 - S4These figures contain too many noninformative boxes. In all the figures, only the last three boxes are informative (reports assessed for eligibility, reports excluded, and studies included in review). The rest boxes convey little information and should be simplified.

We have simplified these diagrams, removing boxes which contained no information.

Figure S6: what does it mean by "prior to the publication of the first article represented in this sample"? What is "this sample"?

“This sample” refers to the collection of 450 GWAS articles, 296 articles using AP-MS, 148 transcriptomics articles, and 15 genome-wide CRISPR screen articles. We have rephrased this sentence to make this clear. It now reads:

“Variant of Figure 1B only considering articles published in 2002 or before, prior to the publication of any of the articles featuring -omics experiments which we considered for this analysis.”